# Immigrant Parents’ Perceived Social Support and Their Children’s Oral Health Behaviors and Caries Experience

**DOI:** 10.3390/ijerph19148250

**Published:** 2022-07-06

**Authors:** Rana Dahlan, Babak Bohlouli, Humam Saltaji, Ida Kornerup, Bukola Salami, Maryam Amin

**Affiliations:** 1Faculty of Medicine and Dentistry, School of Dentistry, University of Alberta, Edmonton, AB T6G 1C9, Canada; dahlan@ualberta.ca (R.D.); saltaji@ualberta.ca (H.S.); kornerup@ualberta.ca (I.K.); 2Department of Dental Public Health, Faculty of Dentistry, King Abdulaziz University, Jeddah 21991, Saudi Arabia; 3Department of Emergency Medicine, University of Alberta, Edmonton, AB T6G 2H1, Canada; bohlouli@ualberta.ca; 4Faculty of Nursing, University of Alberta, Edmonton, AB T6G 2H1, Canada; bukola.salami@ualberta.ca

**Keywords:** social support, oral health, immigrants

## Abstract

This study examined the associations between immigrant parents’ perceived social support (PSS) and their children’s oral health behaviors (OHB) and caries experience. We recruited immigrant parents and children aged 2–12 years. Data were collected on the sociodemographic and OHB information of the children. The parents’ total PSS score and its dimensions were measured using the validated Personal Resource Questionnaire (PRQ2000). Dental examinations determined the children’s caries experience using the DMFT/dmft index. A total of 336 parents and children were included in the study. Household income predicted the parents’ PSS (B = −5.69; 95% CI −9.077, −2.32). Children of parents with higher PSS reported ≥2 toothbrushing/day (*p* ≤ 0.05). Among the PSS domains, parental education level predicted their social integration (B = −0.16; 95% CI −0.30, −0.02) and nurturance (B = −0.24; 95% CI −0.43, −0.06). Family income predicted social integration (B = −0.17; 95% CI −0.33 −0.01), worth (B = −0.23; 95% CI −0.39 −0.06), and assistance (B = −0.22; 95% CI −0.42 −0.01). Parents with higher scores of intimacy and social integration were more aware of their children’s oral health (*p* = 0.01). The parental social integration mean scores were significantly higher among parents whose children consumed ≥1 sugary snack/day (*p* = 0.02). All five domain scores were significantly higher among parents of children who reported ≥2 toothbrushing/day compared with children who brushed <2/day (*p* < 0.05). The results demonstrated that parents’ PSS only improved their children’s toothbrushing frequency. Compared to other domains, social integration was significantly associated with children’s OHB. Neither parental PSS total score nor domains were associated with DMFT/dmft.

## 1. Introduction

Oral health problems are not only the result of biological and behavioral factors; there are also sociocultural-related factors that should be taken into consideration [1,2,3]. Researchers have recently shifted toward understanding the sociocultural factors underlying oral health problems, which are known as the social determinants of oral health. Immigrant populations’ oral health and wellbeing are vulnerable to many changes and challenges after their arrival in host countries, including Canada [4]. Language and cultural barriers, housing and employment problems, low socioeconomic status, and lack of medical and dental coverage are major challenges in many immigrants’ lives [5,6,7].

Immigrants’ oral health can be affected by changes occurring in their social life, such as a reduced number of family members in the host society due to the migration process [8,9,10]. These social ties or connections may affect immigrants’ oral health and children’s oral health in particular [11,12]. Children’s oral health mainly depends on their caregivers and therefore can be affected by the social support their caregivers receive [4,11,12]. Social support can be defined as the support “provided by other people [that] arises within the context of interpersonal relationships” [10,11] and could be provided by formal or informal social groups [13]. Social support may either be perceived or actually received by caregivers. The received social support is the actual support received by individuals [14], while perceived social support is “the individual’s beliefs about the availability of varied types of support from network associates” [14].

The social support received in the host country is vital for immigrants, as it has both direct and indirect effects on their health. Social integration or having strong social ties can directly result in improved self-esteem and satisfaction through some behavioral, emotional, or cognitive means that arise within social relationships and affect health and related behaviors [15,16,17]. Moreover, social support may have a stress-buffering effect, which can indirectly improve immigrants’ health and ease their stress related to illness or post-migration burdens [15,16,17].

Immigrants with high social support have better oral health outcomes, more dental care utilization, and greater knowledge about oral health and dental care [12,18]. As a result, they may have improved oral health quality of life and better self-rated oral health [12,18]. Conversely, immigrants who have less social support are more likely to adopt negative behaviors such as smoking and increased sugar and alcohol consumption that can negatively affect their oral health [12,18].

Caregivers’ perceived or received social support has also been associated with increased dental care utilization, reduced level of dental caries, and the improved oral health behaviors of their children [11,12,18,19]. However, there is limited evidence currently available regarding the impact of immigrant caregivers’ social support on their children’s oral health. Therefore, the aim of this study is to examine the associations between immigrant parents’ perceived social support and their children’s oral health behaviors and caries experience.

## 2. Methods

**Study design and participants:** This cross-sectional quantitative study was conducted in accordance with the Helsinki Declaration, and the results are reported according to STROBE guidelines [20,21]. Ethics approval was obtained from the University of Alberta Research Ethics Board (Protocol # Pro00072345). We included first-generation immigrant parents who had lived in Canada for two years or longer and their children aged 2–12 years.

**Data collection**: The data collection was completed in community locations convenient for the majority of the participants. The recruitment of eligible participants was carried out by multilingual community workers through their existing programs using a non-probability snowball sampling technique. This sampling method allowed us to reach individuals through previously identified community-based organizations and social groups. Information letters were distributed among the participants and informed consent was obtained from parents. The quantitative data were collected from parents through a self-administered questionnaire asking about demographics, their perceived social support, and their children’s oral health behaviors. The clinical data were collected through dental examinations of the children performed by calibrated dentists. The questionnaires were in English. The community workers attended a half-day training session on data collection and assisted with the administration and interpretation of the questionnaires for participants experiencing language barriers.

**Measures**: Parents who gave consent completed a questionnaire consisting of three sections. The first section contained sociodemographic questions and the second section was about their children’s oral health behaviors. In the third section, each parent’s PSS score was measured using the validated Personal Resource Questionnaire (PRQ2000) [22,23,24].

**Outcome variables:** Children’s caries experience and oral health behaviors were the outcome variables of this study. Two calibrated dentists performed dental examinations using a portable dental chair, artificial light, and sterilized mirror and explorer. The World Health Organization-recommended DMFT/dmft index was used for determining the children’s caries experience [25]. The DMFT/dmft is the sum of the number of decayed, missing, and/or filled teeth, with higher scores indicating a higher level of caries experience (World Health Organization, 1997). Children in need of dental treatments were referred to the University of Alberta dental clinic. The children’s oral health behaviors were assessed by an eight-item questionnaire. It included questions on children’s oral health status, last dental visit, toothbrushing frequency, and sugar-intake frequency.

**Independent variable:** The independent variable in this study was parent’s perceived social support, measured using the validated Personal Resource Questionnaire (PRQ2000) [22,23,24]. This scale contains 15 items with five dimensions of support: worth, social integration, intimacy, nurturance, and assistance [22,23,26]. The participants’ responses were recorded on a five-point Likert scale: 1—strongly disagree to 5—strongly agree. The original Likert scale is seven points; however, we changed it to five so that the participants could easily distinguish between the answers. This change was made with permission. The social support score was determined by adding the numbers ranging between 15 and 75, with higher scores indicating a higher level of PSS [22,23,26].

**Covariates:** Based on previous studies [19,27] the covariates included in this study were mother’s age, child’s age, length of residency in Canada, status in Canada (Canadian citizen/permanent resident/temporary resident), parent’s level of education (high school/under, college/trade or university), monthly income (<2000 CAD, 2000–4000 CAD, >4000 CAD), and child dental coverage (yes/no).

**Statistical analysis:** The data were analyzed using the Statistical Package for the Social Sciences (SPSS version 27.0. IBM Corp., Armonk, NY, USA). Discrete variables were reported in percentages, while continuous variables were reported in mean, SD, median, and range, as appropriate. The association between the covariates and PSS was measured using a univariate test. The association between the total PSS score and domains with children’s oral health behaviors and caries experience was measured using two-sample t tests and ANOVA. To investigate the factor structure in our sample, a factor analysis was performed on the PRQ2000 by using the maximum likelihood method of extraction and oblique rotation by the direct oblimin method. Factors were extracted based on their eigenvalues. We then assessed the associations between those extracted factors and children’s oral health-related behaviors and caries experience. Cronbach’s alpha values were calculated to report internal consistency among the items of the PSS scale.

## 3. Results

A total of 336 dyads of one parent and one child were included in this cross-sectional study through 20 community events. The participants were from different communities including South Asians, South East and East Asians, Arabs, Africans, East Europeans, and Hispanics. The majority of the adult participants were mothers with a mean (SD) age of 37 (6.3) years. Around 96% of the mothers were Canadian citizens or permanent residents, with a mean (SD) duration of 8 (5.8) years of living in Canada and only 4% were temporary residents. Regarding income, 42% of the families had a middle income and 67% of the mothers had a college or higher education. With regard to the children, 50.6% were female and 60.7% were born in Canada. Their mean (SD) age was 6.2 (2.8) years (Table 1). 

A total of 48 of the children (48.8%) had visited a dentist within the past 12 months. The reason for their dental visit was a dental checkup in 74% of cases, and only 42% had dental coverage. The mean (SD) DMFT/dmft of the examined children was 3.7 (4.2) in a range of 0–18. About 58% of parents were aware of their child’s dental cavities and 40% rated their child’s oral health as “good”. About 23.5% of children had one or more sugar intakes per day, and 66.1% of them brushed their teeth twice a day (Table 2).

Length of residency in Canada was negatively correlated with the amount of sugar consumption (*p*-value < 0.05). Mother’s age and education, family structure, income level, and dental coverage were significantly associated with increased children’s dental care utilization (*p*-value < 0.05). Toothbrushing frequency was only significantly associated with family structure and mother’s education. Foreign-born children in families with shorter residency lengths in Canada had higher DMFT/dmft scores compared to their native-born peers and families who had lived longer in Canada (*p*-value < 0.05). Bivariate analysis showed that parents’ awareness of their children’s oral health status was associated with higher DMFT/dmft levels compared to parents who were unaware (*p*-value < 0.05) (Table 3).


**Parental Perceived Social Support Total Score:**


The parents’ PSS total score mean (SD) was 63.327 (9.0), with a maximum level of 75. For this sample, the scale had an internal reliability coefficient (Cronbach’s α) of 0.9, which indicates a high level of internal consistency; the scale items were homogeneous (Table 2). 

Length of residency in the host country, current status in Canada, mother’s education, and child’s dental coverage were not significantly associated with parental PSS (*p*-value > 0.05). Family income level was significantly associated with parents’ PSS score; low- (<CAD 2000) and middle-income (CAD 2000–4000) families had lower PSS scores than higher income (>CAD 4000) families (B = −2.82; 95% CI −5.15, −0.49) (Table 4).

The children of parents who perceived a higher level of social support reported toothbrushing twice or more per day compared to brushing less than twice a day (*p*-value ≤ 0.05) (Table 5). The parental PSS mean scores were neither significantly associated with the other oral health behaviors nor with the DMFT/dmft scores. 


**Parental Perceived Social Support Domains:**


The mean (SD) scores for intimacy, social integration, worth, assistance, and nurturance were 12.6 (2.2), 2120 (3.1), 1716 (2.5), 8.4 (1.6), and 4.3 (0.8), respectively (Table 2). 

Social integration and nurturance were significantly associated with parents’ education level. Parents with high school education or less had lower scores of social integration and nurturance than parents with a higher education. Moreover, the social integration, worth, and assistance domains were significantly associated with parents’ income level. Low- (<CAD 2000) and middle-income (CAD 2000–4000) families had lower scores of social integration (B = −0.84; 95% CI −1.65, −0.04), worth (B = −0.90; 95% CI −1.55, −0.26), and assistance (B = −0.43; 95% CI −0.83, −0.02) than higher-income (>CAD 4000) families. However, intimacy was not associated with any demographic variable. None of the domains were significantly associated with length of residency in Canada (Table 4). Mother’s age was positively correlated with the assistance domain (*p*-value = 0.004).

Parents who were aware of their children’s oral health had significantly higher mean (SD) scores of intimacy (12.90 (2.07); *p*-value = 0.01) and social integration (21.33 (3.02); *p*-value = 0.02) compared to parents who were unaware. In addition, the parental social integration mean (SD) scores were significantly higher among children who consumed one or more sugary snacks/day (21.35 (3.20)) compared to children who consumed less than one sugary snack/day (20.57 (2.96); *p*-value = 0.02). All five domain mean scores were significantly higher among the parents of children who reported toothbrushing twice or more a day compared with children who brushed less than twice a day (Table 5). None of the parents’ perceived social support domains were significantly associated with their children’s DMFT/dmft scores.


**Parental Perceived Social Support Factor Analysis:**


In our factor analysis, according to the eigenvalues, scree plot, and parallel analysis, two factors were extracted that explained 57.5% of the total variance. Factor I consisted of eight items and Factor II contained seven items. The factors’ mean (SD) scores were 29.5 (4.6) and 33.8 (5.0) for Factors I and II, respectively. The Cronbach’s alpha was 0.89 for Factor I and 0.88 for Factor II. The inter-item correlations ranged from 0.38 to 0.65 for Factor I and 0.39 to 0.75 for Factor II (Table 2).

Factor I was only significantly associated with parents’ income level, with low- (<CAD 2000) and middle-income (CAD 2000–4000) families having a lower score for that factor (B = −1.60; 95% CI −2.79, −0.40) than higher income (>CAD 4000) families (Table 6). In addition, mother’s age was positively correlated with Factor I (*p*-value = 0.03). Factor II was not associated with any demographic variables. 

Parents who were aware of their children’s oral health reported significantly higher mean (SD) scores of Factor II compared to parents who were unaware: 34.38 (4.84) and 32.94 (5.14), respectively (*p*-value = 0.01). Furthermore, parents of children who consumed one or more sugary snacks/day reported significantly higher mean (SD) scores for Factor II compared to children who consumed less than one sugary snack/day (34.31 (5.04) and 33.15 (4.92), respectively; *p*-value = 0.03). The mean scores of both factors were significantly higher among children who brushed twice or more a day compared with children who brushed less than twice a day (Table 7). Neither Factor I nor Factor II were significantly associated with their children’s DMFT/dmft scores.

**Multivariate logistic regression:** Binominal logistic regression was performed to ascertain the effect of parental perceived social support level and covariates on oral health behaviors. Table 8 presents the results from the three models for the association between PSS and children’s toothbrushing frequency. On average, with a one-unit increase in the PSS score, toothbrushing frequency increased by 5% (OR = 1.05; 95% CI: 1.02–1.06).

## 4. Discussion

Social life changes faced by immigrants after moving to a new country, such as a decreased number of close family members and friends, may lead to isolation and loneliness. This may negatively affect their health and social life due to the adoption of detrimental habits as coping methods [18]. Parental social support within the host country can play a vital role in the oral health outcomes of their children. Therefore, the aim of this study was to measure the parental PSS score and determine whether it had any impact on their children’s oral heath behaviors and caries experience. We hypothesized that children of parents with higher levels of PSS would have improved oral health behaviors and a lower caries experience than children of parents with lower PSS scores. Among all oral health behaviors, only children’s toothbrushing frequency was significantly associated with parental PSS. With regard to demographics, family income level was significantly associated with parental PSS.

In contrast to previous studies [11,12], our results suggest that children of parents with a higher perception of social support had higher toothbrushing frequency. This could be due to an increase in the actual support that parents received, allowing them to gain more information about the importance of promoting positive oral health behaviors [18,28]. In addition, parents who perceive a high level of social support may have a greater ability to extend their relationships not only with friends and family, but also with dental professionals and community workers [18,28]. These individuals may play a vital role in increasing parental awareness and knowledge, which can have a significant impact on their children’s oral health [18,28]. Moreover, children’s toothbrushing frequency was associated with their mother’s level of education in our study. A higher education can result in more awareness of the importance of children’s oral health and toothbrushing frequency. 

On the contrary, we did not find any significant associations between parental PSS score and children’s sugar intake and dental care utilization. There is an inconsistency among the studies examining the associations between parents’ PSS score and their children’s sugar consumption and dental care utilization in the literature. While some studies reported a positive association [19,28,29,30], others’ results were more aligned with our findings [12,31]. However, in our study, children’s sugar intake was positively associated with length of residency in Canada. Longer residency in the host country may lead to higher consumption of a cariogenic diet due to a broad range of factors, including socioeconomic status, neighborhood nature, and post-migration challenges that cause parents to give less priority to their children’s oral health [18,32,33]. 

On the other hand, children’s dental care utilization was linked to other factors, such as income, child’s dental coverage, family structure, mother’s age, and parents’ education. The positive effect of income and dental coverage on dental care utilization may facilitate access to beneficial resources, dental care utilization, and improved oral health behaviors [12,27]. Family structure is another factor influencing a child’s dental care utilization. Children of married parents are more likely to receive dental care than children of single parents [34]. Married parents also had higher levels of social support compared to single parents [34]. It is possible that single parents are more vulnerable to stressors and sole-parenting responsibilities compared to married ones, which could result in a lower rate of dental care utilization [27,34]. In addition, parents who are more educated may be more knowledgeable about oral health and, in turn, would foster more healthy behaviors, including more frequent use of dental services [12]. 

The caries rate was lower among children whose parents had lived in Canada longer. This may be a reflection of their parents’ PSS score as an indirect relation, although this association did not reach the significance level. Social support has been associated positively with length of residency in the host country. In addition, a longer time of residency in the host country allows immigrants to better integrate into mainstream society and receive more benefits from the available social resources, which may result in increased knowledge and greater awareness about their children’s oral health.

In our study, PSS scores were higher among high-income parents than low-income ones, which is consistent with previous studies reporting that low levels of income were associated with a lack of social support [27,35,36]. Stress, depression, and poor social skills are factors associated with post-immigration challenges such as employment and income level [35]. As a result, more limited resources and social support are available to low-income immigrant parents compared to high-income parents who have more access to and receive more support from the surrounding environment [35].

With regard to the parental PSS domains, social integration was the domain most associated with children’s oral behaviors. Parents with high social integration were more aware of their children’s oral health, which consequently led to increased frequency in toothbrushing in their children. This may be a reflection of reduced social strains, improved social network, and perceived social support that results in increased awareness about oral health and better oral health outcomes among their children [37]. On the contrary, increased sugar consumption was reported among children of parents with high social integration. This could be because of a shift from their traditional food to a Western diet, which includes more processed foods, fat, and added sugars [38].

Based on the Canadian Health Measures Survey (CHMS) conducted in 2007–2009, 57% of children aged 6–11 years had a combined dmft and DMFT score of at least 1 [1]. In addition, the average dmft and DMFT was 2.5 for primary or permanent teeth [1]. In our sample, we had 176 children aged 6–12 years with an average DMFT/dmft of 4.28. The average dmft of children aged 2–5 years was also 3.03. With regard to oral health behaviors, the CHMS report indicated that 91.0% of Canadian children had visited a dentist within the previous 12 months, while in our study, only 48.8% reported a dental visit in the year prior to our data collection. These findings confirm a noticeable oral health disparity between children of immigrant populations and their national-born counterparts. 

To the best of our knowledge, this study was the first to adopt the Personal Resource Questionnaire (PRQ2000) (part II) in determining an association between parental PSS and children’s oral health among immigrants. In addition, our study was strengthened by looking at the perceived social support in particular, rather than other types of social support, as PSS is more of a perspective measure. It represents immigrants’ perceptions over a longer duration of time and can enhance adjustment to a new country by promoting positive effects, self-confidence, and a sense of personal satisfaction.

Nonetheless, some limitations of this study need to be acknowledged. First, the cross-sectional design limited our study to only drawing causal inferences and only explained a snapshot of the associations examined. In addition, the implementation of the snowball sampling technique limited the generalizability of our findings, even though it is a widely used method to recruit large samples of minorities and hard-to-reach populations. The self-reported questionnaire is another limitation of our study, as the answers might be biased and more socially desirable. Finally, while our pilot results suggested that we should change the seven-point Likert scale to a five-point one to account for the participants’ literacy level, this change made it difficult for us to compare our findings with those of studies that used a similar instrument, but with the suggested seven-point scale. 

## 5. Conclusions

Immigrant parents’ perceived social support level was associated with higher toothbrushing frequency in their children. However, it was not significantly correlated with either of the other oral health behaviors (i.e., sugar intake and dental visits). Among the five PSS domains, social integration had the strongest association with children’s oral health behaviors. Neither PSS total score nor domains were significantly associated with DMFT/dmft. With regard to the two factors of social support that emerged, Factor II was significantly more associated with oral health behaviors than was Factor I. 

## Figures and Tables

**Table 1 ijerph-19-08250-t001:** Characteristics of participants.

**N of Participants**	**336**
**Mother’s age (mean) (SD)**	37 (6.3)
**Mother’s current status in Canada (N) (%)**	
Temporary residents	13 (3.9)
Permanent residents	183 (54.5)
Canadian citizen	140 (41.7)
**Year of residency in Canada (mean) (SD) (range)**	8 (5.8)
**Household monthly income (N) (%)**	
<CAD 2000	99 (29.5)
CAD 2000–4000	141 (42)
>CAD 4000	96 (28.6)
**Mother’s level of education (N) (%)**	
High school or less	111 (33.0)
Over high school education (college, trade, university, or post graduate)	225 (67.0)
**Childs’ sex (N) (%)**	
Female	170 (50.6)
**Child born in Canada (N) (%)**	204 (60.7)
**Child’s age (Mean) (SD) (Range)**	6.20 (2.8) (2–12)
**Child living with (N) (%)**	
Single parent	45 (13.4)
Both parents	291 (86.6)
**Number of children**	
1	76 (22.6)
2	163 (48.5)
≥3	97 (28.9)

**Table 2 ijerph-19-08250-t002:** Child’s oral health behaviors and parental perceived social support.

**Utilization of Dental Services (Last Year) (N) (%)**	
No	172 (51.2)
Yes	164 (48.8)
**Reason for visit (N) (%)**	
Dental problem	68 (26.0)
Regular dental checkup	193 (74)
**Child with dental coverage (N) (%)**	
Yes	142 (42.3)
**DMFT/dmft (mean) (SD) (range)**	3.68 (4.15) (0–18)
**Parental awareness (N) (%)**	
Not aware (I don’t know)	142 (42.3)
Aware	194 (57.7)
**Parental-rated oral health (N) (%)**	
Poor/fair	132 (39.3)
Good	135 (40.2)
Very good/excellent	69 (20.5)
**Sugar consumption (N) (%)**	
≥1 per day	79 (23.5)
<1 per day	257 (76.5)
**Tooth brushing (N) (%)**	
<twice a day	114 (33.9)
≥twice a day	222 (66.1)
**Starting age for dental brushing by mothers (N) (%)**	
Before age 2	148 (44)
After age 2	188 (56)
**Parental perceived social support (mean) (SD) (range)**	
**Total PSS score**	63.27 (9.0) (34)
**Domains**	
Intimacy	12.6 (2.2) (9)
Social integration	21 (3.1) (12)
Worth	17 (2.5) (13)
Assistance	8.4 (1.6) (6)
Nurturance	4.3 (0.8) (4)
**Factors**	
Factor I	29.5 (4.6) (24)
Factor II	33.8 (5.0) (20)

**Table 3 ijerph-19-08250-t003:** Results of univariate analysis between demographics and children’s oral health behaviors ^Ɨ^.

	Dental Utilization	Reason for Visit	Parents’ Rating	Parents’ Awareness	Brushing Frequency	You Start Brushing	Sugar Consumption
**Child’s gender**	0.38	0.41	0.97	0.32	0.54	0.39	0.80
**Child born in Canada**	0.43	0.69	0.48	0.08	0.51	0.68	0.74
**Parent structure**	0.03 *	0.96	0.81	0.20	0.03 *	0.22	0.33
**Number of children**	0.72	0.44	0.15	0.48	0.59	0.12	0.93
**Current status in Canada**	0.11	0.46	0.40	0.77	0.57	0.62	0.78
**Parents’ education level**	0.00	0.43	0.14	0.80	0.03 *	0.62	0.57
**Income level**	0.003 *	0.64	0.19	0.34	0.10	0.12	0.65
**Dental coverage**	0.05 *	0.23	0.50	0.85	0.23	0.31	0.53

^Ɨ^ Chi-square test for categorical variables; * significant.

**Table 4 ijerph-19-08250-t004:** Association between parental perceived social support level (PRQ2000) and demographics (b coefficient, 95% CI, and *p*-values).

Demographics	Parental Perceived Social Support (PRQ85)		Intimacy	Social Integration	Worth	Assistance	Nurturance
**Length of residence (years)**	−0.01 (−0.18, 0.16) 0.89		−0.02, (−0.06, 0.02) 0.41	−0.01 (−0.07, 0.05) 0.76	0.01 (−0.03, 0.06) 0.57	0.01 (−0.02, 0.04) 0.67	−0.01 (−0.02, 0.01) 0.48
**Current status in Canada ^Ɨ^**	**PR**	0.30, (−4.41, 6.16) 0.77	0.09 (−0.40, 0.58) 0.73	0.01 (−0.68, 0.70) 0.97	0.08 (−0.48, 0.64) 0.78	−0.02 (−0.37, 0.33) 0.92	0.14 (−0.04, 0.32) 0.21
**Parents’ level of education**	−1.72 (−3.77, 0.33) 0.10		−0.22 (−0.73, 0.28) 0.38	−0.80 (−1.50, −0.10) 0.02 *	−0.34 (−0.91,0.23) 0.24	−0.11 (−0.47, 0.25) 0.54	−0.24 (−0.43, −0.06) 0.02 *
**Income level ^ƗƗ^**	**A**	−1.06 (−3.59, 1.46) 0.41	−0.09 (−0.70, 0.53) 0.79	−0.26 (−1.13, 0.62) 0.57	−0.48 (−1.18,0.22) 0.18	−0.22 (−0.66, −0.02) 0.32	−0.02 (−0.25, 0.21) 0.86
	**B**	−2.82 (−5.15, −0.49) 0.02 *	−0.44 (−1.02, 0.13) 0.13	−0.84 (−1.65, −0.04) 0.04 *	−0.90 (−1.55, −0.26) 0.01 *	−0.43 (−0.83, −0.02) 0.04 *	−0.20 (−0.42, 0.01) 0.06
**Child has dental coverage**	0.78 (−1.18, 2.74) 0.43		0.27 (−0.22, 0.75) 0.28	0.13 (−0.55, 0.81) 0.71	0.11 (−0.44, 0.66) 0.70	0.30 (−0.04, 0.64) 0.08	−0.02 (−0.20, 0.16) 0.80

^Ɨ^ Canadian citizens; ^ƗƗ^ more than CAD 5000; * significant; A: <CAD 2000; B: CAD 2000–4000; PR: permanent residents.

**Table 5 ijerph-19-08250-t005:** Parental perceived social support domains’ (PRQ2000) mean score difference by children’s oral health behaviors and DMFT/dmft.

		N	Social Support (Mean, SD)	*p*-Value	Intimacy	*p*-Value	Social Integration	*p*-Value	Worth	*p*-Value	Assistance	*p*-Value	Nurturance	*p*-Value
**Utilization of dental services (last year)**	No	172	4.22, 0.81	0.76	12.61, 2.32	0.8	21.09, 3.29	0.50	17.02, 2.51	0.71	8.38, 1.65	0.80	4.28, 0.81	0.47
	Yes	164	61.12, 8.39		12.67, 2.09		20.87, 2.91		16.92, 2.51		8.43, 1.49		4.22, 0.833	
**Reason for visit**	Dental problem	68	63.84, 8.55	0.66	12.83, 2.16	0.50	21.16, 3.09	0.70	17.04, 2.72	0.81	8.42, 1.58	0.76	4.37, 0.80	0.20
	Regular checkup	193	63.30, 8.55		12.63, 5.19		20.99, 3.01		16.95, 2.39		8.49, 1.49		4.22, 0.80	
**Parents’ rating**	Poor/fair			0.42		0.33		0.63		0.58		0.31		0.169
	Good/very good													
	Excellent													
**Parents’ awareness**	Unaware	142	62.20, 9.34	0.06	12.28, 2.34	0.01 *	20.51, 3.18	0.02 *	16.90, 2.53	0.63	8.32, 1.64	0.39	4.18, 0.86	0.18
	Aware	194	64.06, 8.70		12.90, 2.07		21.33, 3.02		17.03, 2.50		8.47, 1.52		4.30, 0.79	
**Brushing frequency**	<2 times	114	60.48, 8.99	<0.001 *	12.17, 2.11	0.005 *	20.18, 2.92	<0.001 *	16.07, 2.75	<0.001 *	8.02, 1.58	<0.001 *	8.02, 1.59	<0.001 *
	≥2 times	222	64.71, 8.70		12.89, 2.22		21.40, 3.12		17.44, 2.24		8.60, 1.53		8.60, 1.53	
**Start cleaning**	After 2 years old	148	62.93	0.47	12.58, 2.15	0.65	20.95, 3.17	0.85	16.80, 2.64	0.26	8.34, 1.62	0.50	4.25, 0.79	0.95
	Before 2 years old	188	63.54, 8.76		12.69, 2.26		21.02, 3.08		188, 17.12, 2.40		8.46, 1.54		4.26, 0.855	
**Sugar consumption**	≥1/day	180		0.816	12.79, 2.19	0.19	21.35, 3.20	0.02 *	17.21, 2.45	0.07	8.48, 1.56	0.40	4.30, 0.77	0.26
	<1/day	156			12.47, 2.22		20.57, 2.96		156, 16.71		156, 8.33, 1.60		4.20, 0.88	
**DMFT/dmft**				0.40		0.86		0.24		0.51		0.68		0.17

* significant.

**Table 6 ijerph-19-08250-t006:** Association between parental perceived social support factors (PRQ2000) and demographics (B coefficient, 95% CI and *p*-values).

Demographics		Factor 1	Factor 2
**Length of residence (years)**		0.01 (−0.07, 0.10) 0.80	−0.02 (−0.12, 0.07) 0.62
**Current status in Canada ^Ɨ^**	**PR**	−0.07 (−1.09, 0.96) 0.90	0.37 (−0.74, 1.48) 0.51
**Parents’ level of education**		−0.60 (−1.66, 0.45) 0.26	−1.11 (−2.25, 0.03) 0.06
**Income level ^ƗƗ^**	**A**	−0.61 (−1.90, 0.69) 0.36	−0.46 (−1.87, 0.95) 0.52
	**B**	−1.60 (−2.79, −0.40) 0.01 *	−1.22 (−2.53, 0.08) 0.07
**Child has dental coverage**		0.54 (−0.47, 1.54) 0.29	0.24 (−0.85, 1.33) 0.66

^Ɨ^ Canadian citizens; ^ƗƗ^ more than CAD 5000; * significant; A: <CAD 2000; B: CAD 2000–4000.

**Table 7 ijerph-19-08250-t007:** Parental perceived social support (PRQ2000) factors’ mean score difference by children’s oral health behaviors and DMFT/dmft.

		N	Factor 1 (Mean, SD)	*p*-Value	Factor 2 (Mean, SD)	*p*-Value
**Utilization of dental services** **(last year)**	No	172	29.48, 4.86	0.92	33.94, 5.23	0.53
	Yes	164	29.53, 4.39		33.59, 4.78	
**Reason for visit**	Dental problem	68	29,56, 4.88	0.84	34.28, 5.13	0.34
	Regular checkup	193	29.69, 4.31		33.61, 4.89	
**Parents’ rating**	Poor/fair	132		0.47		0.47
	Good/very good	135				
	Excellent	69				
**Parents’ awareness**	Unaware	142	29.27, 43,80	0.42	32.94, 5.14	0.01
	Aware	194	29.68, 4.51		34.38, 4.84	
**Brushing frequency**	<2 times	114	28.13, 4.89	<0.001	32.35, 4.96	<0.001
	≥2 times	222	30.21, 4.34		34.50, 4.89	
**Start cleaning**	After 2 years old	148	29.25, 4.80	0.37	33.68, 5.08	0.78
	Before 2 years old	188	29.71, 4.49		33.84, 4.97	
**Sugar consumption**	≥1/day	180	29.82, 4.53	0.18	34.31, 5.04	0.03
	<1/day	156	29.14, 4.73		33.15, 4.92	
**DMFT/dmft**				0.80		0.19

**Table 8 ijerph-19-08250-t008:** Results of logistic regression analysis for the association between parents’ perceived social support level and children’s brushing frequency (*n* = 336; OR (95% CI); *p*-value).

	Model 1	Mode 2	Model 3
**Parents’ perceived social support level**	1.06 (1.02,1.06) <0.001 *	1.05 (1.02, 1.06) <0.001 *	1.05 (1.02, 1.06) <0.001 *
**Child born in Canada ^Ɨ^**	0.79 (0.46, 1.37) 0.30	0.73 (0.42, 1.29) 0.21	0.741 (0.42, 1.31) 0.23
**Years in Canada**	1.01 (0.96, 1.06) 0.49	1.02 (0.97,1.08) 0.28	1.02 (0.97, 1.07) 0.33
**Current status in Canada** ** ^ƗƗ^ ** **Permanent residents**	0.79 (0.45, 1.37) 0.30	0.72 (0.45, 1.16) 0.21	0.72 (0.44, 1.16) 0.20
**Parents’ level of education** ** ^ƗƗƗ^ ** **High school or less**		0.61 (0.35, 1.06)0.09	0.61 (0.35, 1.06) 0.8
**Income level** **^ƗƗƗƗ^**			
**<CAD 2000**		1.42 (0.72, 2.83) 0.40	1.55 (0.77, 3.11) 0.30
**CAD 2000–4000**		0.84 (0.47, 1.51) 0.42	0.89 (0.49, 1.62) 0.54
**With dental coverage ^ƗƗƗƗƗ^**			1.41 (0.86, 2.31) 0.18

^Ɨ^ No; ^ƗƗ^ Canadian citizens; ^ƗƗƗ^ college, university, trade, or post grad; ^ƗƗƗƗ^ <CAD 5000; ^ƗƗƗƗƗ^ no; * significant.

## Data Availability

The supporting generated data during the study can be found at the University of Alberta School of Dentistry, Edmonton, AB, Canada.

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
