# Peer review of "Immigrant Parents’ Perceived Social Support and Their Children’s Oral Health Behaviors and Caries Experience"

_ijerph, 2022, doi:10.3390/ijerph19148250_

Round 1
Reviewer 1 Report
The title should be revised to convey what this research is about.
Please define in your manuscript's title what children's oral health studied.
Please write the purpose of this study in the past tense (in the Abstract). The Abstract says you examined oral health behaviors and caries experience.
Please specify whether you collected data from an adult as a proxy for a child, or an adult himself/herself.
Please specify the number of children, number of adults, and who are they (mother, father, legal guardian, foster parent, etc.). You wrote: "Parents with higher PSS reported ≥2 toothbrushing/day (P≤0.05)." Is this related to them or their children? It is not clear.
Many of your study participants are Canadian citizens (or even born in Canada). It is not clear what are the criteria that you recruited your participants.
Please define these based on their immigration status (and where they are originally from).
Also, please clarify the difference in expected findings between your sample population and Canadian whether they are Indigenous or non-Indigenous.
Author Response
"Please see the attachment."

Reviewer 2 Report
The topic is interesting, but the data is presented in a confusing way.
The same value is sometimes presented to two decimal places, sometimes rounded to one decimal place.
There are errors in reference to the table in the text.
The tables must be better paginated to be clearer.
In table 3 "income level": explain what it's A and B.
in Table 2 one of the range value is missing.
In my opinion the results must be totally rewritten
Author Response
"Please see the attachment."

Round 2
Reviewer 1 Report
The authors have addressed the comments during my first round of review. However, the note concerning the type of immigration and originally from where the immigrants had come was not adequately addressed. Please specify if the immigrants were skilled workers, refugees, ...etc, as well as their home countries. Another note concerning the study findings and their comparison with Indigenous people (who are also national born) was not included in the authors' revisions.
